# From symptom to cancer diagnosis: Perspectives of patients and family members in Alberta, Canada

Anna Pujadas Botey[1,2]*, Paula J. Robson[2,3], Adam M. Hardwicke-Brown[3], Dorothy M. Rodehutskors[4], Barbara M. O'Neill[3], Douglas A. Stewart[1,5]

1 Cancer Strategic Clinical Network, Alberta Health Services, Calgary, Alberta, Canada, 2 School of Public Health, University of Alberta, Edmonton, Alberta, Canada, 3 Cancer Strategic Clinical Network, Alberta Health Services, Edmonton, Alberta, Canada, 4 CancerControl Alberta, Alberta Health Services, Calgary, Alberta, Canada, 5 Departments of Oncology and Medicine, University of Calgary, Calgary, Alberta, Canada

* Anna.PujadasBotey@ahs.ca

**Data Availability Statement:** All relevant data are within the paper. Further data excerpts are available on request from the corresponding author.

**Funding:** The authors received no specific funding for this work.

## Abstract

### Background

Significant intervals from the identification of suspicious symptoms to a definitive diagnosis of cancer are common. Streamlining pathways to diagnosis may increase survival, quality of life post-treatment, and patient experience. Discussions of pathways to diagnosis from the perspective of patients and family members are crucial to advancing cancer diagnosis.

### Aim

To examine the perspectives of a group of patients with cancer and family members in Alberta, Canada, on factors associated with timelines to diagnosis and overall experience.

### Methods

A qualitative approach was used. In-depth, semi-structured interviews with patients with cancer (n = 18) and patient relatives (n = 5) were conducted and subjected to a thematic analysis.

### Findings

Participants struggled emotionally in the diagnostic period. Relevant to their experience were: potentially avoidable delays, concerns about health status, and misunderstood investigation process. Participants emphasized the importance of their active involvement in the care process, and had unmet supportive care needs.

### Conclusion

Psychosocial supports available to potential cancer patients and their families are minimal, and may be important for improved experiences before diagnosis. Access to other patients' lived experiences with the diagnostic process and with cancer, and an enhanced supportive

**Competing interests:** The authors have declared that no competing interests exist.

role of family doctors might help improve experiences for patients and families in the interval before receiving a diagnosis of cancer, which may have a significant impact on wellbeing.

## Introduction

Cancer is the leading cause of death in Canada [1]. About half of Canadians will develop cancer in their lifetime, and about one-fourth will die of the disease [2]. Evidence suggests that diagnosing cancer at earlier stages may be associated with improvements in survival [3].

Significant intervals from first noticing a symptom to receiving a cancer diagnosis (known as the diagnostic period) have been widely documented in the literature [4, 5]. A number of researchers have reported associations between longer intervals and later stages of cancer at diagnosis, reduced survival, decreased quality of life post-treatment, and suboptimal patient experience [3, 6]. In Alberta, analyses of administrative data spanning 2004–2011 described potentially preventable long periods from suspicion to diagnosis for breast, colorectal, and lung cancers [7–10]. In Canada and across the world there are substantial variations in the length of the diagnostic period for different cancers [4, 11], and numerous studies have focused on understanding factors that may influence this timeline in order to improve health outcomes and patient experience [12].

The importance of discussing the path to diagnosis from the perspective of patients and family members is increasingly acknowledged in the literature [13]. Receiving a cancer diagnosis is often preceded by a period of waiting for a diagnosis following the discovery of symptoms, which is anxiety-provoking [14]. The psychosocial impacts of the wait may be accentuated if patients believe there have been inefficiencies in their pathway to diagnosis. In Canada, discussions to advance cancer diagnosis are mostly based on timeline-related information available in administrative databases [9, 10], and not so much on perceived timelines and patient and family experiences [6, 15]. This study was designed to contribute to filling that gap. The objective was to examine the perspectives of a group of patients and family members in Alberta, Canada, on factors associated with timelines to diagnosis and overall patient and family experience. Alberta has a publicly-funded, provincially operated single-payer healthcare system (Alberta Health Services), in which all residents have free access to standard medical care. Primary care, diagnostic and specialized services are covered by public funding. Patients may also, if that is their preference and can afford it, use certain diagnostic services provided in private clinics. The results of this study will be of interest to health services' researchers, clinicians, operational leads and decision-makers involved in optimizing the care of potential cancer patients, including those working in the areas of primary care, diagnostic services, and cancer care. Learning more about patients and family members' perceptions and experiences related to the diagnostic period may help inform improvements in health system organization and the development of interventions to minimize stress and improve satisfaction with care, which may have a significant impact on wellbeing [16].

## Methods

### Participants

Participants were associated with the Patient & Family Advisor Network (PFAN) of Cancer-Control Alberta (Alberta Health Services). PFAN is a community of people who are committed to using their experiences to help improve the health system. Participants in this study

were patients with cancer or relatives of patients with cancer who had received a cancer diagnosis within the last three years, and were living in Alberta at the time of diagnosis.

To recruit participants, the PFAN coordinator sent all PFAN members an electronic engagement request inviting expressions of interest in participating in the study. The PFAN coordinator then sent interested members an email with information about the study and a screening questionnaire for them to complete. Forty two PFAN members completed the questionnaire. They were cancer patients and family members, who represented different types of cancer, sex and age ranges, treatment types (curative or non-curative), and residence (rural and urban). Responses to the questionnaire were used to purposefully select participants [17]. The first selection technique used was intensity sampling, which involves the selection of information-rich cases to ensure a variety of experiences and perspectives [17]. We initially selected four participants representing each of the three most common types of cancer (breast, lung and colorectal): two male and two female participants related to each cancer type, with equal representation of rural and urban residence among genders and cancer types. Two family relatives of two of the initially selected participants were willing to participate and were included as part of the initial pool of participants. After conducting and analyzing these initial interviews, we concluded that more interviews were required to reach data saturation, meaning that new themes (or concepts) emerged as we analyzed further interviews. From the original list of participants related to the three cancer types described earlier, we randomly selected and interviewed two additional participants before reaching saturation. From that point we wanted to deepen further the themes emerging from the analysis. We used a maximum variation sampling technique to select additional participants representing cases that varied from each other [17]. In the analysis we identified that cancer type was a relevant dimension of variation, and using variation sampling we selected participants related to diverse cancer types not yet represented in our sample. We invited further participants until no new insights emerged from additional interviews. The selection of participants was made by the first author and the PFAN coordinator. The PFAN coordinator invited selected participants by sending them an email that included a consent form to be reviewed prior to the interview. In total, 23 individuals participated in the study (18 patients and five family members), and 20 interviews were conducted (17 with one participant, and three with two participants).

## Procedure

The framework used for the study was 'Model of Pathways to Treatment' [18–20]. It identifies four intervals from suspicion of a health problem to receiving treatment: 1) from detection of symptoms to perceiving a reason to discuss symptoms with a healthcare provider (*appraisal*); 2) from perceiving a reason to discuss symptoms with a healthcare provider to first consultation (*help-seeking*); 3) from first consultation to formal diagnosis, including the initiation of investigation, prescription of tests, examinations, and diagnosis (*diagnosis*); and, 4) from formal diagnosis to start of treatment (*pre-treatment*) [19]. The study covers the first three intervals, referred to collectively as the diagnostic period.

This qualitative study followed a phenomenological approach [21]. Phenomenological studies examine and describe phenomena as they are consciously experienced by individuals [22]. This approach allowed us to fully investigate the diagnostic period from the lived experiences of patients with cancer and their families, and their shared meanings of these experiences [23, 24]. The purpose was to capture new insights that may inform how to improve experiences during cancer diagnosis, complementing the previous quantitative approaches undertaken in Alberta. In-depth, semi-structured interviews were used for data collection. Interviews followed an interview guide that was developed in close collaboration with PFAN leadership and

with feedback from patient advisors affiliated with the Cancer Strategic Clinical Network (S1 and S2 Appendices). Using accepted qualitative research standards [23], pilot interviews were conducted with four patients with cancer to ensure the interview guide answered the proposed research objective. The interview guide covered topics such as how participants made sense of their symptoms, why they chose to visit a healthcare provider and how they experienced going through appointments, referrals, and tests before they were provided with a definitive cancer diagnosis. It also included a section on recommendations for improvement including the need for emotional support during the diagnostic period.

The study was conducted with written ethics approval granted by the Health Research Ethics Board of Alberta–Cancer Committee (Study ID: HREBA.CC-18-0210). Interviews were conducted by APB, a qualitative researcher by background, with a PhD in social sciences, with interest in the diagnostic period and no previous work in the area with patients or family relatives. She was involved in the early development of the study, and had no prior relationship or sharing of personal information with the participants approached for interview. Interviews were conducted face-to-face at a time and location convenient for each participant. Most participants were interviewed in a room at their closest cancer centre, and three in a meeting room in the facility where the PFAN coordinator worked. In one case it was not possible to find a convenient location and the interview was conducted through videoconference. There was no presence of non-participants during the interviews. Before proceeding with each interview, participants were invited to sign the consent form that they had received from the PFAN coordinator by email. Interviews took place between June 26 and September 7, 2018, and lasted an average of 41 minutes (range 29–88 minutes). During each interview the researcher took field notes to maintain contextual details and non-verbal expressions. All interviews were audio-recorded and subsequently transcribed verbatim. To protect the identity of participants, at the transcription stage each interview was assigned an identification number and any identifying information was deleted.

## Analysis

Data analysis was done concurrently with data collection. All interviews were transcribed verbatim, and interview transcripts were imported into NVivo Version 11 (QSR International, Australia). Transcripts were thematically analyzed using an inductive data-driven coding process to reflect on how participants made meaning of their experiences without predetermined theories [21, 25]. This process entailed a methodical review of the full text of each interview transcript. It began with close readings of transcripts and consideration of the multiple meanings that were inherent in the text. The researcher then identified text segments that contained meaning units, and created a code (or label) for each new theme into which the text segment was assigned. Additional text segments were assigned to each code where they were relevant. As the review of the text progressed, existing codes were revised, new codes were created, and more text segments were assigned to the codes. Later, the researcher developed initial descriptions of the meaning of the codes, and according to these meanings they linked codes to other codes in various relations such as causal sequences or a hierarchy of codes (S3 Appendix). The researcher who conducted the interviews did all of the coding. To ensure consistency and trustworthiness [25], a second researcher coded randomly-selected segments of text. The two researchers discussed their interpretation and codes until they reached consensus. An interrater reliability analysis using Cohen's Kappa was performed for 20% of the coded interviews. The Kappa value was 0.81, which can be interpreted as "strong agreement" [26]. To increase validity, participants were sent research findings and given the opportunity to provide feedback (by email or phone), and findings and feedback were validated in a subsequent group discussion with eight patients with cancer [25].

## Results

Patients with cancer and family members who participated in the study represented ten different cancers, 74% of them were women (n = 17), their median age was 59 years (range 42–76), 70% resided in urban locations (n = 16), 65% of patients had been treated with curative intent (n = 15), and the median time between the date of diagnosis and the date of the interview was 19 months (range 2–36). Participants had diverse experiences during the diagnostic period, but some commonalities exist. Thematic analysis revealed three salient themes as being relevant to their experience (Fig 1): potentially avoidable delays, concerns about health status, and misunderstood investigation process. Participants struggled emotionally in the period between identifying symptoms and receiving a cancer diagnosis, and had suboptimal care experiences.

### Potentially avoidable delays

Participants referred to potentially avoidable long periods of time spent in the diagnostic period. They mentioned delays related to the patient, to the doctor, and to the health system.

**Patient-related delays.** Initial inaction by patients caused delays. Patients did not initially act because they did not identify symptoms or did not think symptoms were signs of a serious problem. An additional cause of delay was that patients postponed visits with their doctor due to fear of cancer, being busy, feeling embarrassed or not feeling it was an appropriate use of the doctor's time.

Having some degree of awareness about the seriousness of symptoms, and knowing individuals who had experienced cancer played a very important role in acknowledging the problem and deciding to go to the doctor promptly after noticing a symptom. As this participant

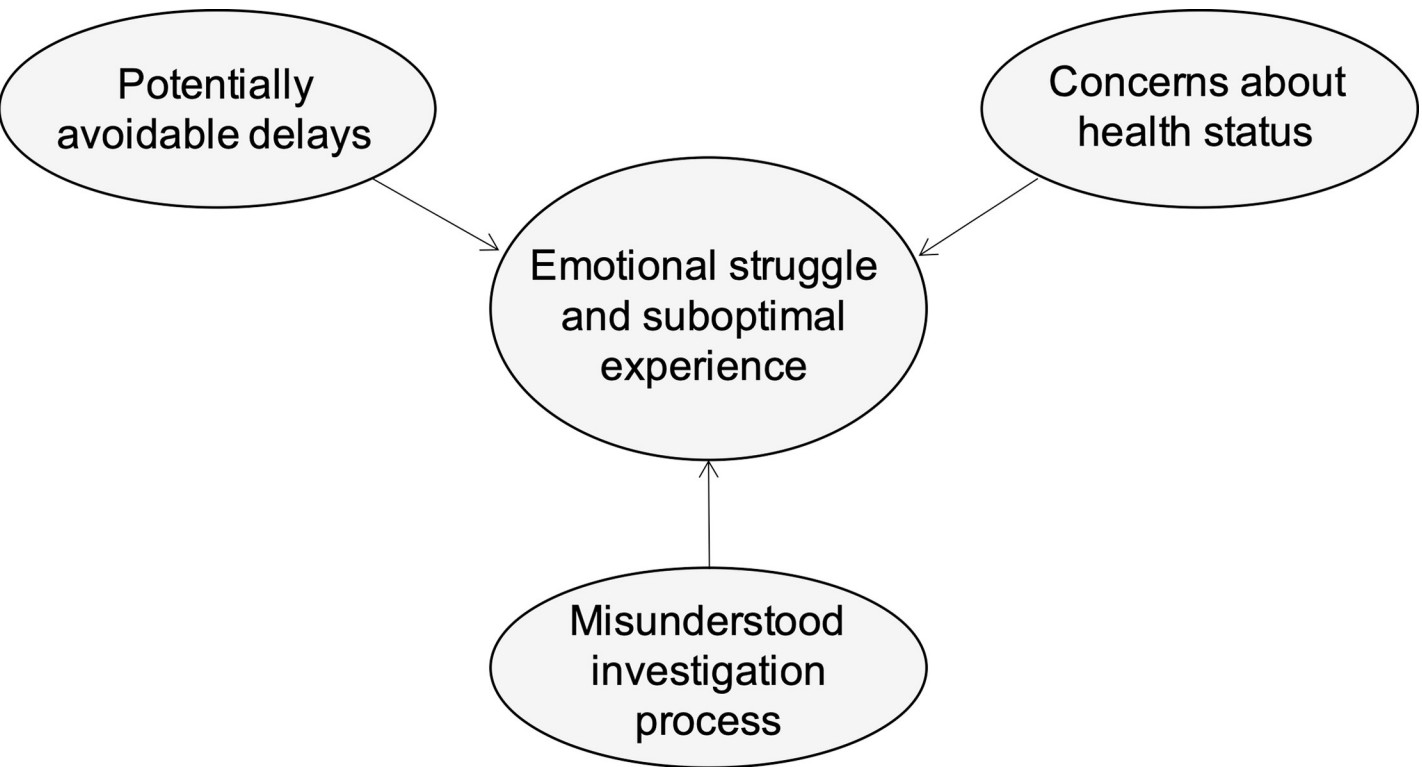

**Fig 1. Emergent themes relevant to cancer patients and family members' experience during the diagnostic period.**

explained it: "A friend from the community had [breast cancer] and died of it. So, I felt this lump. As soon as we got home, I made an appointment" [breast cancer patient 10].

**Family doctor-related delays.** In some cases, participants reported that they believed that their doctor's inability to identify symptoms led to patients undergoing investigation for diseases other than cancer, or a later visit to the emergency department. One participant explained this delay in eventually investigating and identifying cancer as: "she just developed this cough. It wouldn't go away. We took her to the doctor and just said it's allergies [. . .]. Several months went by, and she wasn't any better. We went to the doctor again and again before even thinking of cancer" [relative of lung and brain cancer patient 12].

Participants perceived family doctors as the patients' doorway to the diagnostic pathway. In the context of their experience, some participants referred to their family doctor as the person who restricted access to the investigation of cancer symptoms and jeopardized a timely diagnosis. One explained: "I guess it couldn't be cancer if [the doctor] didn't even check for it" [breast cancer patient 2], and:

> "It isn't until [doctors] are convinced something is wrong, nothing is going to happen [. . .]. It was not until we did the private MRI that we found out [she had cancer]. That triggered the doctor really believing that something was wrong and doing something about her cancer" [relative of lung cancer patient 3].

In some cases, participants debated about the role of family doctors and agreed that doctors could play a bigger role at supporting patients in navigating the healthcare system. One participant, reflecting on her experience, said: "I've had this family doctor for many years [. . .]. I felt that he should have been more involved in getting things moving quicker" [colon cancer patient 14].

**System-related delays.** Long waiting times to see a specialist, undergo testing or learn test results delayed diagnosis. Some participants spent extensive time following the multiple steps involved in the investigation. They shared frustration with what they reflected to be an uncoordinated and inefficient process. A few shared that there was not much they could have done about it, while others expressed that "there's a responsibility for [patients] to be more proactive" [ovarian cancer patient 4]. As explained by this participant:

> "In my case, people doing the test didn't follow through. So, that process was dropped. But I still had those symptoms, so we should have done more investigation. I had to be assertive enough to say to the doctor: 'I need this exam'" [colon cancer patient 15].

Being familiar with cancer and having system connections or being knowledgeable about how the system works were mentioned as enablers for self-advocacy during the diagnostic period and prompt diagnosis. One participant explained, for example:

> "I was phoning and trying not to harass, but I knew what my options were, what I can do. I remember phoning my family doctor and, you know, 'can you get us in?' to get a colonoscopy earlier. Unless you are an advocate, you get lost in the system" [relative of lung cancer patient 2].

Many participants commented on their long wait to receive test results. In particular, they shared their frustration with the long wait for the results to be available after testing, and to get an appointment (or phone call in some cases) with their family doctor to discuss test results. For example, this participant said: "waiting three weeks [after a mammogram] was really

stressful. Not getting the appointment sooner. Waiting for that to happen, and you just feel pretty powerless because there is nothing you can do" [breast cancer patient 1]. None of the participants spoke to the situation in which they were left waiting with an expectation that no news is good news, and some of the comments participants made suggest that this would not be a typical situation; "you are just not kept waiting. People are just not waiting for the bad news, so if you do not see the doc or they don't call you, you try to call or see somebody" [anal cancer patient 18]. Rather, most participants were advised to book an appointment with their family doctor at the time of diagnostic testing to discuss results, and given a timeframe for the results to be available; "[they told me:] you should get your results within a week so book an appointment with your GP for the next week" [breast cancer patient 1].

## Concerns about health status

Participants mentioned that it was stressful to consider the possibility of a major health condition such as cancer. In most cases, family members were mentioned as the primary source of emotional support. Some participants also assigned an emotional supporting role to family doctors: "they could be the hub, supporting the patient" [prostate cancer patient 5]. In terms of the timing of health concerns, most participants mentioned that before the investigation "you want to trivialize it, it's not a big deal" [breast cancer patient 1], but once the investigation process started, they began to consider the possibility of a serious health problem.

Participants shared three elements that triggered being concerned about a serious health problem. First, the fact that their family doctor decided to pursue testing or referral to a specialist prompted worries. "I thought: 'nobody was too worried about it until now, why do I have to go [to the specialist] suddenly?' So I became worried" [anal cancer patient 18]. Second, the realization that the investigation was not as straightforward as they thought caused concerns. "It was scary; Dr. [name] sent us for blood tests again, and the scans, that was like two days" [relative of prostate cancer patient 5]. Third, the existence or non-existence of communication with healthcare providers during the investigation caused patients to worry about their health. "I was having the mammogram, the radiologist was there. She showed me what she saw on the screen and she said: 'this is what I am looking at', and I'm concerned about that" [breast cancer patient 17]. "A lot of cancer people say that at the radiology place, when they are doing [a mammogram], the radiologist often comes in and explains; they can often tell right away. But nothing, that did not happen, and I'm worried" [breast cancer patient 1].

## Misunderstood investigation process

The process of investigation was described differently by participants who accessed care through the emergency department and those who accessed care through their family doctor. Participants who went to the emergency department described the investigation as "very abrupt, instantaneous and with no stress, because it was right away" [breast cancer patient 2]. In several cases, participants who saw their doctor and had the investigation initiated by their doctor explained that the investigation was cumbersome and stressful. They referred to specific complications such as having tests done multiple times or having to travel to inconvenient locations for medical appointments, and also to the fact that it involved multiple steps that were often unexpected and seemed uncoordinated. As explained by this participant: "I just thought I was gonna go in for a mammogram [. . .]. I went for my mammogram and I thought everything would be fine, but they called me back and they said: 'we need to see you again'" [breast cancer patient 10]. What seemed particularly stressful for participants was the fact they did not understand what the investigation process entailed, and were unsure about what to expect. As explained by one participant:

"There didn't seem to be a plan, really. I mean, I know they have a plan because that's what they've done a million times before, but they don't share that too much other than to say: 'we are going to do a biopsy or whatever test'. There is not a whole lot of explanation [. . .]. I was just following what they were saying, and I was: 'ok, I'll just show up there for that appointment and do that'" [breast cancer patient 16].

After reflecting on how challenging it had been for them to navigate the investigation process, some participants stressed that others might struggle. In particular, they referred to patients from cultural minorities, elderly, with less education or other characteristics that may impact their ability to advocate for themselves. "Where am I supposed to go? People don't have a clue; [. . .] you feel lost, it's very scary", and "I kept saying: 'if you were an immigrant coming in here, how would you ever figure all this out?'" [anal cancer patient 18]. Participants emphasized the importance of having family members who can support patients, and the need to improve the coordination of services and have resources available to patients: "whether it be a nurse navigator, whether it be some written information, or like just somebody to tell you: 'this is what [the investigation process] looks like and these are the timelines'" [breast cancer patient 6].

## Discussion

The study described the diagnostic period from the perspective of a group of patients with cancer and family members in Alberta. Findings contribute to the literature by focusing on perceived challenges along the pathway to diagnosis, and thus may inform improvements in health system organization and have implications for the development of interventions to improve the experiences of patients and family members. Of relevance, findings showed that patients and family members participating in the study experienced anxiety and had suboptimal experiences. Participants expressed the importance of self-advocacy and wanting to have an active role in their care, and suggested unmet supportive care needs when navigating the system. The study made two novel contributions to the literature that require additional investigation: 1) the role of accessing patients' experiences with cancer in helping symptom recognition and dealing with a potential cancer diagnosis, and 2) the importance of an active role of patients and families in their care before receiving a cancer diagnosis. Overall, findings from the study highlight a need for further discussion on the provision of psychosocial supports to potential cancer patients and their families before they have a confirmed diagnosis of cancer.

The finding of participants feeling distressed during this period is consistent with previous studies undertaken in various countries including the United States, Denmark and Canada [13, 18]. Before diagnosis, individuals with symptoms suspicious of cancer face uncertainty and threat of a serious illness while having to undergo medical appointments and tests, which might be fearsome, uncomfortable and demanding. Findings from the study are also in alignment with the literature by indicating that fear about having a serious health condition is an important cause of anxiety among patients and family members waiting for diagnosis [27]. Also in accordance with the literature, findings showed that perceived long timelines add stress to an already stressful situation [28]. Consistent with previous research, participants associated delays with their inaction or late action in seeking medical attention [6], with their doctor's failure to correctly identify cancer symptoms [12, 29], and with system inefficiencies including variable access to specialists and testing and limited coordination of care [30, 31]. Of relevance, participants in the study referred to cancer awareness and second-hand experience with cancer as two important elements supporting the patient's prompt consult with a doctor. While the relevance of cancer awareness has been previously acknowledged [32, 33], the influence of

shared cancer stories on the experiences of potential cancer patients with the diagnostic period has not been discussed in the literature, to the best of our knowledge. In the post-diagnosis literature, access to patients' real experiences with the disease has been reported important in providing general support to patients with cancer including the provision of information related to cancer and care process, and emotional help [34, 35]. The potential role of having access to these experiences in not just helping individuals seek medical attention, but also dealing with information and emotional aspects of a prospective cancer diagnosis at the beginning of their journey requires further investigation.

A relevant contribution from our study is the importance of self-advocacy and the need to better support patients and family members acquire an active role in their care prior to receiving a cancer diagnosis. Similar to what has been documented in the post-diagnosis literature [36], participants in the study referred to the concept of 'patient activation' [37]. They articulated their need or willingness and ability to take independent actions to manage their health and care throughout the diagnostic period. Participants did not only refer to their responsibility for seeking medical attention after identifying symptoms potentially related to cancer, but also for getting their family doctor to facilitate access to the investigation of these symptoms, and for finding their way through the investigation process. On more than one occasion, participants shared that in order to avoid delays and decrease their anxiety they had opted to take on tasks that are typically handled by their doctors such as choosing a particular test, or getting prompt specialist appointments. Similar to what has been discussed in the context of treatment and survivorship [38], the healthcare system, even in the pre-diagnosis period, is increasingly complex, with a growing number and variety of specialties, care delivery sites, and diagnostic tests. In the face of navigating these complexities, and in the context of the psychological burdens of potentially facing cancer it is not surprising that many patients and families find it crucial to advocate for getting their care needs met [35]. In order to fully benefit from the care available to them and to enhance have improved experiences, patients and families spoke to the need to promote their own interests and actively try to avoid situations such as delayed appointments with a specialist or delayed access to appropriate testing.

As acknowledged by participants and supported by previous research, in order to be able to advocate for themselves, patients and family members need to be involved with individuals and groups that support their interests [35, 36]. Unlike post-diagnosis, before diagnosis patients are not "cancer patients" yet and they and their family members do not hold membership in particular patient/family groups that can provide this support. Once a cancer diagnosis has been made, oncology care teams are an important source of support to patients and families [39]. However, before diagnosis, patients often expect this role to be played by family doctors. Family doctors in Canada play a key role in helping to manage and coordinate care for patients before and after diagnosis [40], and as suggested by our findings their engagement from the beginning of the patients' journey in helping them navigate the diagnostic period and advocating for them when required is important. We heard from participants that it is important for patients/families to feel that family doctors are on their side and support them. Family doctors can support patient and family members willing to engage in self-advocacy behaviours by trying to establish deeper connections with them and adequately informing them. As reported in previous research, family doctors are a primary source of information during the diagnostic period [15], and patients need to feel adequately informed in order for them to participate effectively in making decisions throughout the diagnostic period [35]. Based on our findings, optimal attention to patient and family needs may include hearing their concerns, providing them with regular updates on the investigation process, engaging in discussions of expectations about the different steps involved in the diagnostic period, and assessing their understanding of these conversations. Another important consideration, particularly relevant

in the Canadian context, is the diversity of cultural groups, whose perspectives and values about health and wellbeing may differ from the dominant culture [41]. For example, First Nations, Inuit and Métis Peoples in Canada have cultural worldviews and definitions of health that are not reflected in the dominant biomedical approach to health care, and as result they may face important challenges to accessing and using health services [42, 43]. Respect and understanding of these different views, including diverse expectations and responsibilities of patients and healthcare providers, are crucial for appropriate care delivery. Finding ways to promote and support family doctors in the care of all potential cancer patients throughout the diagnostic period and equipping them with the skills and tools required to better support them should be encouraged and explored further.

One of the concerns over widening the traditional scope of healthcare providers to better support patients in taking a more active role in their care is health literacy and the demands placed on patients with low literacy. Health literacy refers to the social and cognitive skills and abilities of individuals to gain access to, understand and use information to make appropriate health decisions [44]. Low health literacy is associated with health inequalities, and people with low health literacy tend to have poorer health and use an increased amount of healthcare resources [45]. As a more active role of patients is gaining importance [46], the boundaries of health literacy are being extended beyond the healthcare system to all sectors of life [47]. Encouraging individuals to take control of their health may result in having people better equipped to navigate the complex healthcare system, and also has the potential to optimize health throughout life [48]. Improving health literacy may be a relevant element in the ongoing evolution towards a stronger, more accessible and equitable health care system.

## Limitations

The findings of this study should be considered in light of some limitations. First, the study was not longitudinal and results are particular to the time frame the interviews were conducted. Distress and recovery from the diagnostic period is a dynamic process, the perspectives of patients and family members would likely be different at an earlier or later point in time. Their perceptions may have changed as more time elapsed since the date of diagnosis. Second, the information was collected retrospectively and data might be subject to memory recall errors. Third, as in any study involving self-reported data, results may be subject to exaggeration and attribution. Finally, it is important to acknowledge the implications of the participant selection method employed. While purposefully sampling allows the selection of cases (or types of cases) that can best enhance the researcher's understanding of the phenomenon under study, chosen cases may have particular perceptions that could impact reaching data saturation across other thematic areas, and have a potential effect on the findings. Despite these limitations, results from this exploratory study provide important contributions to this area given the limited number of studies focused on perceived timelines and challenges during the diagnostic period.

## Conclusion

In an effort to streamline the diagnostic period in Alberta, Alberta Health Services and other stakeholders have established streamlined diagnostic programs for breast, lung and prostate cancers [49–51]. The Cancer Strategic Clinical Network is currently initiating a program to establish an Alberta Facilitated Cancer Diagnosis Strategy that goes across cancer sites and geographies in the province. To help inform this work, our study asked patients and family members in the province about their experiences from symptom to cancer diagnosis and found that they struggled emotionally. Findings complement quantitative studies that

described long and variable periods from suspicion to diagnosis for different cancers [8–10]. Findings suggested that from the perspectives of patients and family members, shorter times to diagnosis are desirable, but many additional factors also need to be considered for improved satisfaction with care. Findings further suggested that it would be beneficial to design particular interventions targeting individuals dealing with a prospective cancer diagnosis, as well as healthcare providers, with emphasis on family doctors. Ideally, these interventions should be co-designed with all stakeholders. Approaches to be considered include increasing awareness of cancer, providing peer-support from others who have had experiences with the disease and the diagnostic process, and facilitating access to the types of supports currently available to people who have received a cancer diagnosis. In the case of health care providers, interventions to increase awareness of patients' experiences and their needs for support during the diagnostic process, as well as interventions for enhancing their abilities and skills in supporting patients, may be pertinent. An important factor to be considered when developing and implementing these strategies is the acknowledgment of the particular context of each patient/family, and their specific needs throughout the whole process. Although this study helps shed light on how to improve the experiences of patients and family members during the diagnostic period, more work is required with multiple stakeholders to understand how to better support them throughout this difficult time.

## Supporting information

**S1 Appendix. Semi-structured interview protocol (patients).**
(DOCX)

**S2 Appendix. Semi-structured interview protocol (family members).**
(DOCX)

**S3 Appendix. Coding example.**
(DOCX)

**S1 Fig. Emergent themes relevant to cancer patients and family members' experience during the diagnostic period.**
(TIF)

## Acknowledgments

The authors acknowledge the Patient & Family Advisor Network (CancerControl Alberta) for their assistance in recruiting participants, and feedback on data collection protocol; Debora Allatt, Lorelee Marin, Frances Cusano, and Charlie Fischer for their feedback; and study participants for their insights.

## Author Contributions

**Conceptualization:** Anna Pujadas Botey, Paula J. Robson, Adam M. Hardwicke-Brown, Dorothy M. Rodehutskors, Barbara M. O'Neill, Douglas A. Stewart.

**Formal analysis:** Anna Pujadas Botey.

**Investigation:** Anna Pujadas Botey, Dorothy M. Rodehutskors.

**Methodology:** Anna Pujadas Botey, Paula J. Robson, Adam M. Hardwicke-Brown, Dorothy M. Rodehutskors.

**Project administration:** Anna Pujadas Botey.

**Supervision:** Anna Pujadas Botey.

**Visualization:** Paula J. Robson.

**Writing – original draft:** Anna Pujadas Botey.

**Writing – review & editing:** Anna Pujadas Botey, Paula J. Robson, Adam M. Hardwicke-Brown, Dorothy M. Rodehutskors, Barbara M. O'Neill, Douglas A. Stewart.

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
