## [Decision Letter · Decision Letter 0]

9 Jun 2020

PONE-D-20-12776

From symptom to cancer diagnosis: Perspectives of patients and family members in Alberta, Canada

PLOS ONE

Dear Dr. Pujadas Botey,

Thank you for submitting your manuscript to PLOS ONE. After careful consideration, we feel that it has merit but does not fully meet PLOS ONE’s publication criteria as it currently stands. Therefore, we invite you to submit a revised version of the manuscript that addresses the points raised during the review process.

We look forward to receiving your revised manuscript.

Kind regards,

Alvaro Galli

Academic Editor

PLOS ONE

Journal Requirements:

2. Please provide details of the obtained participant consent in the ethics statement on the online submission form. Currently this information is only available in the methods section of your manuscript.

Additional Editor Comments (if provided):

Reviewers' comments:

Reviewer's Responses to Questions

**Comments to the Author**

1. Is the manuscript technically sound, and do the data support the conclusions?

Reviewer #1: Yes

Reviewer #2: Yes

2. Has the statistical analysis been performed appropriately and rigorously? 

Reviewer #1: I Don't Know

Reviewer #2: N/A

3. Have the authors made all data underlying the findings in their manuscript fully available?

Reviewer #1: No

Reviewer #2: Yes

4. Is the manuscript presented in an intelligible fashion and written in standard English?

Reviewer #1: Yes

Reviewer #2: Yes

5. Review Comments to the Author

Reviewer #1: Overall, the study is very well written and contributes to the existing literature. However the novelty of new findings is not clearly stated. Furthermore, the methods are not completely clear. Some considerations listed below:

1. There are more participants than interviews – which suggests that some interviews may have been done with multiple participants at the same time, although this is not indicated

2. Although topic area very important, the overall scope of the study seems fairly limited

3. Details about how the contextual factors of the Canadian health system (or more specifically, the health system in Alberta) may have contributed to the findings is minimally discussed thus generalizability of findings is uncertain

4. Although purposeful sampling is a strength, it may also be a limitation given that various groups of individuals may have varied perceptions which could impact reaching saturation across other thematic areas

5. The article is descriptive, yet with this qualitative work, the authors do not outline a plan for next steps – based on this data, what types of interventions could be proposed/developed; and what other factors need to be considered with intervention development.

Reviewer #2: Thank you for the opportunity to review your paper. The diagnostic period can indeed be stressful for patients and families as they cope with uncertainty and anxiety while waiting for a definitive diagnosis. The period between tests and diagnosis can be one of deafening silence, which is apparent in your participant interview comments.

I have made a few comments which are embedded on the attached document. I would like to see a better description and explanation of qualitative research method used and data analyses process. The paper would benefit with some further details about the need for self-advocacy. It is not clear if patients were provided a timeframe for results and if there were delays in this; I wondered how many were simply told that their GP would follow up with them and then proceeded to wait with an expectation that no news is good news. Was consideration given to health literacy and health beliefs in this as some patients have different beliefs about whose responsibility things are around health and wellbeing. Cultural contexts?

6. PLOS authors have the option to publish the peer review history of their article (what does this mean?). If published, this will include your full peer review and any attached files.

Reviewer #1: No

Reviewer #2: No

---

## [Author Response · Author response to Decision Letter 0]

9 Jul 2020

Manuscript PONE-D-20-12776

Response to Reviewers 

Dear Dr. Galli, 

Thank you for giving us the opportunity to submit a revised draft of the manuscript “From symptom to cancer diagnosis: Perspectives of patients and family members in Alberta, Canada” for publication in PLOS ONE. We appreciate the time and effort that you and the reviewers dedicated to providing feedback on our manuscript and are grateful for the insightful comments on and valuable improvements to our paper. We have incorporated all the suggestions made by the reviewers. Those changes are highlighted within the manuscript. Please see below, in blue, for a point-by-point response to the reviewers’ comments and concerns. All line numbers refer to the revised manuscript file with tracked changes.

Reviewer #1

Comment 1. Overall, the study is very well written and contributes to the existing literature. However the novelty of new findings is not clearly stated. Furthermore, the methods are not completely clear. 

Authors’ Response. Thank you for this comment. In response to the second part, we have added a sentence clearly stating the novel contributions of this work to the literature at the beginning of the Discussion section (line numbers 321-324):

The study made two novel contributions to the literature that require additional investigation: 1) the role of socializing patients’ experiences with cancer in helping symptom recognition and dealing with a potential cancer diagnosis, and 2) the importance of an active role of patients and families in their care before receiving a cancer diagnosis.

In response to the third part, we have made substantial edits to the methods to provide further details and enhance clarity. Details on these edits are provided below, in our response to Comments 2, 5, 7.2, 7.3, 7.4, and 8. 

Comment 2. There are more participants than interviews – which suggests that some interviews may have been done with multiple participants at the same time, although this is not indicated.

Authors’ response. Thank you for pointing out the need to clarify this. We have added a statement that explains that 17 interviews were conducted with one participant and three were conducted with two participants. We conducted a total of 20 interviews and had a total of 23 participants. These edits were made where we were originally presenting the total number of interviews conducted (line numbers 113-116).

Comment 3. Although topic area very important, the overall scope of the study seems fairly limited.

Authors’ response. We agree that the overall scope of the study is modest. However, we believe that the strengths of the study in its conceptualization, design and execution contribute to the delineation of the issue of long timelines from first symptoms to cancer diagnosis. The study illuminates this issue identified in the literature from the perspectives of patients and family members, which has been paid limited attention and is crucial for advancing cancer diagnosis. The study makes important and novel contributions to the literature, and may importantly inform improvements in the health system organization and the development of interventions to improve the experiences of cancer patients when navigating the diagnostic period. Practical contributions are applicable to Alberta and other jurisdictions with similar healthcare systems.

To clarify and further emphasize the potential impact of the study we have edited the last section of the introduction (line numbers 71-77) to say:

The results of this study will be of interest to health services’ researchers, clinicians, operational leads and decision-makers involved in optimizing the care of potential cancer patients, including those working in the areas of primary care, diagnostic services, and cancer care. Learning more about patients and family members’ perceptions and experiences related to the diagnostic period may help inform improvements in health system organization and the development of interventions to minimize stress and improve satisfaction with care, which may have a significant impact on wellbeing [1]. 

Comment 4. Details about how the contextual factors of the Canadian health system (or more specifically, the health system in Alberta) may have contributed to the findings is minimally discussed thus generalizability of findings is uncertain

Authors’ response. We think this is an excellent suggestion. We have now included a description of the health system in Alberta (line numbers 68-71):

Alberta has a publicly-funded, provincially operated single-payer healthcare system (Alberta Health Services), in which all residents have free access to standard medical care. Primary care, diagnostic and specialized services are covered by public funding. Patients may also, if that is their preference and can afford it, use certain diagnostic services provided in private clinics.

This addition facilitates a better understanding of the research objective, findings, discussion and potential contributions of the study.

Comment 5. Although purposeful sampling is a strength, it may also be a limitation given that various groups of individuals may have varied perceptions which could impact reaching saturation across other thematic areas

Authors’ response. We agree with the reviewer’s comment. Accordingly, we have included a statement in the Limitations section of the manuscript acknowledging the potential impact that purposeful sampling could have on reaching saturation (line numbers 418-423):

Finally, it is important to acknowledge the implications of the participant selection method employed. While purposefully sampling allows the selection of cases (or types of cases) that can best enhance the researcher’s understanding of the phenomenon under study, chosen cases may have particular perceptions that could impact reaching data saturation across other thematic areas, and have a potential effect on the findings. 

Comment 6. The article is descriptive, yet with this qualitative work, the authors do not outline a plan for next steps – based on this data, what types of interventions could be proposed/developed; and what other factors need to be considered with intervention development.

Authors’ response. We have edited part of the conclusion section to clearly present the particular interventions to be proposed based on our findings, as well as factors to be considered when developing (and implementing) these interventions (line numbers 435-454):

Findings suggested that from the perspectives of patients and family members, shorter times to diagnosis are desirable, but many additional factors also need to be considered for improved satisfaction with care. Findings further suggested that it would be beneficial to design particular interventions targeting individuals dealing with a prospective cancer diagnosis, as well as healthcare providers, with emphasis on family doctors. Ideally, these interventions should be co-designed with all stakeholders. Approaches to be considered include increasing awareness of cancer, providing peer-support from others who have had experiences with the disease and the diagnostic process, and facilitating access to the types of supports currently available to people who have received a cancer diagnosis. In the case of health care providers, interventions to increase awareness of patients’ experiences and their needs for support during the diagnostic process, as well as interventions for enhancing their abilities and skills in supporting patients, may be pertinent. An important factor to be considered when developing and implementing these strategies is the acknowledgment of the particular context of each patient/family, and their specific needs throughout the whole process. Although this study helps shed light on how to improve the experiences of patients and family members during the diagnostic period, more work is required with multiple stakeholders to understand how to better support them throughout this difficult time. 

Reviewer #2

Thank you for the opportunity to review your paper. The diagnostic period can indeed be stressful for patients and families as they cope with uncertainty and anxiety while waiting for a definitive diagnosis. The period between tests and diagnosis can be one of deafening silence, which is apparent in your participant interview comments.

Comment 7. I have made a few comments which are embedded on the attached document.

Comment 7.1. First sentence, second paragraph, Introduction: Rephrasing needed - this sentence isn't grammatically correct and doesn't make sense as written; suggest splitting into two. In the second part of this statement, the intent is not clear as the first part refers to symptom onset and receiving a diagnosis. What is the next part about ...related to later stages at diagnosis...?

Authors’ response. Thank you for pointing this out. We have edited this sentence. As suggested, we have split it in two sentences, and reworded the second part. The new wording in the second part clearly presents the implications of long intervals from first noticing a symptom to receiving cancer diagnosis (which was the initial intent of the sentence). The revised text reads as follows (line numbers 49-53):

Significant intervals from first noticing a symptom to receiving a cancer diagnosis (known as the diagnostic period) have been widely documented in the literature [2, 3]. A number of researchers have reported associations between longer intervals and later stages of cancer at diagnosis, reduced survival, decreased quality of life post-treatment, and suboptimal patient experience [4, 5].

Comment 7.2. Second sentence, second paragraph, Participants: Expand on this process please - how many were selected initially (given the data saturation comment) based on characteristics as described? Were there so many with each type of cancer (e.g. breast, colorectal, etc.; so many per rural/urban living situation)? 

Authors’ response. We have substantially expanded this section to provide a more comprehensive explanation of the participant selection process. The manuscript now describes the particular techniques we used to select participants, and we have added a description of the different steps followed until reaching saturation. We have also included a description of the characteristics of the participants selected at the initial and subsequent steps. 

The revised paragraph read as follows (line numbers 90-113):

Forty two PFAN members completed the questionnaire. They were cancer patients and family members, who represented different types of cancer, sex and age ranges, treatment types (curative or non-curative), and residence (rural and urban). Responses to the questionnaire were used to purposefully select participants [6]. The first selection technique used was intensity sampling, which involves the selection of information-rich cases to ensure a variety of experiences and perspectives [6]. We initially selected four participants representing each of the three most common types of cancer (breast, lung and colorectal): two male and two female participants related to each cancer type, with equal representation of rural and urban residence among genders and cancer types. Two family relatives of two of the initially selected participants were willing to participate and were included as part of the initial pool of participants. After conducting and analyzing these initial interviews, we concluded that more interviews were required to reach data saturation, meaning that new themes (or concepts) emerged as we analyzed further interviews. From the original list of participants related to the three cancer types described earlier, we randomly selected and interviewed two additional participants before reaching saturation. From that point we wanted to deepen further the themes emerging from the analysis. We used a maximum variation sampling technique to select additional participants representing cases that varied from each other [6]. In the analysis we identified that cancer type was a relevant dimension of variation, and using variation sampling we selected participants related to diverse cancer types not yet represented in our sample. We invited further participants until no new insights emerged from additional interviews. The selection of participants was made by the first author and the PFAN coordinator. The PFAN coordinator invited selected participants by sending them an email that included a consent form to be reviewed prior to the interview. 

We believe that these additions do not only clarify the participant selection process, but also help better explain the analysis process, and how it is intertwined with the process of selecting participants and interpreting what they shared in the interviews.

Comment 7.3. First sentence, second paragraph, Procedure: This is a pretty broad description of qualitative research. Did your team use a specific approach to qualitative research? e.g. grounded theory, phenomenology? The data were collected using in-depth interviews but please expand on your methodological approach to the research beyond 'qualitative approach'. 

Authors’ response. We have added the suggested content to the manuscript. Phenomenology is the methodological approach we used for the study. An explanation of this approach as well as the rationale for choosing it has been included (line numbers 127-131):

This qualitative study followed a phenomenological approach [7]. Phenomenological studies examine and describe phenomena as they are consciously experienced by individuals [8]. This approach allowed us to fully investigate the diagnostic period from the lived experiences of patients with cancer and their families, and their shared meanings of these experiences [9, 10].

Comment 7.4. Analysis: An illustrative example of the coding would be helpful in the appendices.

Authors’ response. As suggested, we have included an example of inductive coding to illustrate the coding process in the appendices (Appendix S3). In particular, this example shows how specific text segments in one of the interview transcripts were examined for meaning and assigned to emerging codes, and how these codes were described and linked to other codes in a causal sequence and hierarchy. We believe this example brings additional clarity to the Analysis section.

Comment 7.5. First sentence, third paragraph, Discussion: This is a really important point and I wonder if consideration is made for health literacy in relation to one's health? Also consider cultural implications about health and wellbeing. 

Authors’ response. We agree about the importance of this point, and appreciate this comment. Health literacy, and cultural implications of health and wellbeing are relevant concepts, and have now been included in the discussion. In particular, we have made two additions to the manuscript:

1) We have added a statement in the text discussing how healthcare providers can better support patients by highlighting the importance of considering the implications of particular cultural contexts (line numbers 388-395):

Another important consideration, particularly relevant in the Canadian context, is the diversity of cultural groups, whose perspectives and values about health and wellbeing may differ from the dominant culture [11]. For example, First Nations, Inuit and Métis Peoples in Canada have cultural worldviews and definitions of health that are not reflected in the dominant biomedical approach to health care, and as result they may face important challenges to accessing and using health services [12, 13]. Respect and understanding of these different views, including diverse expectations and responsibilities of patients and healthcare providers, are crucial for appropriate care delivery. 

2) We have written a new paragraph that augments this discussion by emphasizing the concept of health literacy and arguing how improving health literacy may be an important element in the ongoing evolution towards a stronger, more accessible and equitable health care system (line numbers 400-410). It reads as follows:

One of the concerns over widening the traditional scope of healthcare providers to better support patients in taking a more active role in their care is health literacy and the demands placed on patients with low literacy. Health literacy refers to the social and cognitive skills and abilities of individuals to gain access to, understand and use information to make appropriate health decisions [14]. Low health literacy is associated with health inequalities, and people with low health literacy tend to have poorer health and use an increased amount of healthcare resources [15]. As a more active role of patients is gaining importance [16], the boundaries of health literacy are being extended beyond the healthcare system to all sectors of life [17]. Encouraging individuals to take control of their health may result in having people better equipped to navigate the complex healthcare system, and also has the potential to optimize health throughout life [18]. Improving health literacy may be a relevant element in the ongoing evolution towards a stronger, more accessible and equitable health care system.

Comment 8. I would like to see a better description and explanation of qualitative research method used and data analyses process.

Authors’ response. We agree that the method and data analysis process could be better described and explained. We have made a number of changes to the manuscript that address this comment. Frist, we have specified the methodological approach used in the study (phenomenology) and included an explanation about it (see response to Comment 7.3). Second, we have substantially expanded on the participant selection process, and clarified the number of interviews conducted with one and two participants (see response to Comments 7.2 and Comment 2–Reviewer #1). Third, we have provided further details on the analysis process including a clear description of the coding process, which has been further clarified by the addition of an appendix presenting a coding example (see response to Comment 7.4).

The revised text describing the analysis process (line numbers 161-174) reads as follows: 

Data analysis was done concurrently with data collection. All interviews were transcribed verbatim, and interview transcripts were imported into NVivo Version 11 (QSR International, Australia). Transcripts were thematically analyzed using an inductive data-driven coding process to reflect on how participants made meaning of their experiences without predetermined theories [7, 19]. This process entailed a methodical review of the full text of each interview transcript. It began with close readings of transcripts and consideration of the multiple meanings that were inherent in the text. The researcher then identified text segments that contained meaning units, and created a code (or label) for each new theme into which the text segment was assigned. Additional text segments were assigned to each code where they were relevant. As the review of the text progressed, existing codes were revised, new codes were created, and more text segments were assigned to the codes. Later, the researcher developed initial descriptions of the meaning of the codes, and according to these meanings they linked codes to other codes in various relations such as causal sequences or a hierarchy of codes (S3 Appendix). 

Comment 9. The paper would benefit with some further details about the need for self-advocacy.

Authors’ response. We have added the suggested content to the manuscript in the Discussion section (line numbers 355-360):

Participants did not only refer to their responsibility for seeking medical attention after identifying symptoms potentially related to cancer, but also for getting their family doctor to facilitate access to the investigation of these symptoms, and for finding their way through the investigation process. On more than one occasion, participants shared that in order to avoid delays and decrease their anxiety they had opted to take on tasks that are typically handled by their doctors such as choosing a particular test, or getting prompt specialist appointments.

Comment 10. It is not clear if patients were provided a timeframe for results and if there were delays in this; I wondered how many were simply told that their GP would follow up with them and then proceeded to wait with an expectation that no news is good news.

Authors’ response. We agree that this information was not clear, and have added a further explanation about it. We went back to the interview transcripts and can confirm that none of the participants spoke to the situation in which they were left waiting with an expectation that no news is good news. In addition, comments made suggest that this would not be a typical situation. Rather, most participants were advised to book an appointment with their family doctor at the time of diagnostic testing to discuss results, and given a timeframe for the results to be available.

The added piece of text reads as follows (line numbers 248-260):

Many participants commented on their long wait to receive test results. In particular, they shared their frustration with the long wait for the results to be available after testing, and to get an appointment (or phone call in some cases) with their family doctor to discuss test results. For example, this participant said: “waiting three weeks [after a mammogram] was really stressful. Not getting the appointment sooner. Waiting for that to happen, and you just feel pretty powerless because there is nothing you can do” [Breast cancer patient 1]. None of the participants spoke to the situation in which they were left waiting with an expectation that no news is good news, and some of the comments participants made suggest that this would not be a typical situation; “you are just not kept waiting. People are just not waiting for the bad news, so if you do not see the doc or they don’t call you, you try to call or see somebody” [Anal cancer patient 18]. Rather, most participants were advised to book an appointment with their family doctor at the time of diagnostic testing to discuss results, and given a timeframe for the results to be available; “[they told me:] you should get your results within a week so book an appointment with your GP for the next week” [Breast cancer patient 1]. 

Comment 11. Was consideration given to health literacy and health beliefs in this as some patients have different beliefs about whose responsibility things are around health and wellbeing. Cultural contexts?

Authors’ response. This comment has been addressed together with Comment 7.5. Details provided in response to that comment. 

References:

1. Jonikas J, Grey D, Copeland M, Razzano L, Hamilton M, Floyd C, et al. Improving propensity for patient self-advocacy through wellness recovery action planning: Results of a randomized controlled trial. Community Ment Health J. 2013;49(3):260-9. doi: 10.1007/s10597-011-9475-9. PubMed PMID: 104198141. Language: English. Entry Date: 20130731. Revision Date: 20150711. Publication Type: Journal Article.

2. Butler J, Foot C, Bomb M, Hiom S, Coleman M, Bryant H, et al. The International Cancer Benchmarking Partnership: An international collaboration to inform cancer policy in Australia, Canada, Denmark, Norway, Sweden and the United Kingdom. Health Policy. 2013;112(1-2):148-55. Epub 2013/05/23. doi: 10.1016/j.healthpol.2013.03.021. PubMed PMID: 23693117.

3. Maringe C, Walters S, Rachet B, Butler J, Fields T, Finan P, et al. Stage at diagnosis and colorectal cancer survival in six high-income countries: A population-based study of patients diagnosed during 2000-2007. Acta Oncol. 2013;52(5):919-32. Epub 2013/04/16. doi: 10.3109/0284186x.2013.764008. PubMed PMID: 23581611.

4. Brousselle A, Breton M, Benhadj L, Tremblay D, Provost S, Roberge D, et al. Explaining time elapsed prior to cancer diagnosis: Patients' perspectives. BMC Health Serv Res. 2017;17:448. doi: 10.1186/s12913-017-2390-1. PubMed PMID: WOS:000405236700001.

5. Neal RD, Tharmanathan P, France B, Din NU, Cotton S, Fallon-Ferguson J, et al. Is increased time to diagnosis and treatment in symptomatic cancer associated with poorer outcomes? Systematic review. Br J Cancer. 2015;112:S92-S107. doi: 10.1038/bjc.2015.48. PubMed PMID: WOS:000355372300014.

6. Patton MQ. Qualitative research & evaluation methods. Integrating theory and practice. Fourth ed. Saint Paul, MN: SAGE Publications, Inc; 2015.

7. Denzin N, Lincoln Y. The SAGE handbook of qualitative research 5th ed. Thousand Oaks, CA: SAGE Publications, Inc; 2018.

8. Moustakas C. Phenomenological research methods. Thousand Oaks, CA: SAGE Publications, Inc.; 1994.

9. Creswell J. Research design: Qualitative, quantitative, and mixed methods approaches: Thousand Oaks: Sage Publications; 2014.

10. Van Manen M. Phenomenology of practice : meaning-giving methods in phenomenological research and writing: Left Coast Press; 2014.

11. Rootman I, Ronson B. Literacy and health research in Canada: Where have we been and where should we go? Canadian Journal of Public Health. 2005;96(2):62-7. PubMed PMID: edsjsr.41994460.

12. Waldram JB, Adelson N, Lipinski A, Fiske J-A, Fletcher C, Denham A, et al. Aboriginal healing in Canada: Studies in therapeutic meaning and practice: Aboriginal Healing Foundation; 2008.

13. Allen L, Hatala A, Ijaz S, Courchene ED, Bushie EB. Indigenous-led health care partnerships in Canada. CMAJ. 2020;192(9):E208-E16. doi: 10.1503/cmaj.190728. PubMed PMID: 142065133. Language: English. Entry Date: In Process. Revision Date: 20200321. Publication Type: journal article. Journal Subset: Biomedical.

14. Kickbusch I, Pelikan J, Apfel F, Tsouros A. Health literacy: The solid facts. Copenhagen: World Health Organization, 2013.

15. Institute of Medicine, Committee on Health Literacy, Nielsen-Bohlman L, Panzer AM, Kindig DA. Health literacy: A prescription to end confusion. Washington DC: National Academies Press; 2004.

16. Coulter A, Parsons S, Askhan J. Policy Brief: Where are the patients in decision-making about their own care? Copenhagen: World Health Organization, 2008.

17. Huber JT, Shapiro RM, Gillaspy ML. Top down versus bottom up: The social construction of the health literacy movement. The Library Quarterly: Information, Community, Policy. 2012;82(4):429-51. doi: 10.1086/667438. PubMed PMID: 79892467.

18. Kickbusch I, Maag D, Saan H. Enabling healthy choices in modern health societies. European Health Forum Badgastein 2005, 5–8 October 2005. 2005.

19. Miles MB, Huberman MA, Saldana J. Qualitative data analysis: A methods sourcebook. 3rd ed. Thousand Oaks, CA: SAGE Publications, Inc; 2014.

---

## [Decision Letter · Decision Letter 1]

18 Aug 2020

PONE-D-20-12776R1

From symptom to cancer diagnosis: Perspectives of patients and family members in Alberta, Canada

PLOS ONE

Dear Drs. Pujadas Botey

Thank you for submitting your manuscript to PLOS ONE. After careful consideration, we feel that it has merit but does not fully meet PLOS ONE’s publication criteria as it currently stands. Therefore, we invite you to submit a revised version of the manuscript that addresses the points raised during the review process.

Please address the comments raised by reviewers 3 and submit your revised manuscript by Oct 02 2020 11:59PM. If you will need more time than this to complete your revisions, please reply to this message or contact the journal office at plosone@plos.org. Please include the following items when submitting your revised manuscript:

We look forward to receiving your revised manuscript.

Kind regards,

Alvaro Galli

Academic Editor

PLOS ONE

Reviewers' comments:

Reviewer's Responses to Questions

**Comments to the Author**

1. If the authors have adequately addressed your comments raised in a previous round of review and you feel that this manuscript is now acceptable for publication, you may indicate that here to bypass the “Comments to the Author” section, enter your conflict of interest statement in the “Confidential to Editor” section, and submit your "Accept" recommendation.

Reviewer #3: All comments have been addressed

Reviewer #4: All comments have been addressed

2. Is the manuscript technically sound, and do the data support the conclusions?

Reviewer #3: Yes

Reviewer #4: Yes

3. Has the statistical analysis been performed appropriately and rigorously? 

Reviewer #3: N/A

Reviewer #4: Yes

4. Have the authors made all data underlying the findings in their manuscript fully available?

Reviewer #3: No

Reviewer #4: Yes

5. Is the manuscript presented in an intelligible fashion and written in standard English?

Reviewer #3: Yes

Reviewer #4: Yes

6. Review Comments to the Author

Reviewer #3: I did not review this manuscript in the first round.

However, it appears that the authors have been responsive to all of the comments of the original reviewers.

I only have one technical comment. In the description of trustworthiness (Methods), the authors say that a second investigator coded a random sample of segments and that disagreements were settled via consensus. However, this tells us little about how often two skilled coders would come to similar conclusions. Either present information on reliability or list this as a limitation.

Also, at the bottom of page 13, the sentence that starts "the role of socializing patients' experiences with cancer..." does not make sense to me. What does it mean to "socialize" experiences? Please clarify or cut.

Reviewer #4: The authors have done an excellent job in responding to reviewer comments and have improved the readership of the manuscript. This is an important topic for improving quality of life for patients with cancer.

7. PLOS authors have the option to publish the peer review history of their article (what does this mean?). If published, this will include your full peer review and any attached files.

Reviewer #3: No

Reviewer #4: No

---

## [Author Response · Author response to Decision Letter 1]

27 Aug 2020

Manuscript PONE-D-20-12776

Response to Reviewers 

Dear Dr. Galli, 

Thank you for giving us the opportunity to submit a revised draft of the manuscript “From symptom to cancer diagnosis: Perspectives of patients and family members in Alberta, Canada” for publication in PLOS ONE. We appreciate the time and effort that you and the reviewers dedicated to providing feedback on our manuscript and are grateful for the insightful comments on and valuable improvements to our paper. We have incorporated all the suggestions made by the Reviewer #3. Those changes are highlighted within the manuscript. Please see below, in blue, for a point-by-point response to their comments. All line numbers refer to the revised manuscript file with tracked changes.

Reviewer #3

Comment 1. I did not review this manuscript in the first round. However, it appears that the authors have been responsive to all of the comments of the original reviewers.

Authors’ response. Thank you.

Comment 2. I only have one technical comment. In the description of trustworthiness (Methods), the authors say that a second investigator coded a random sample of segments and that disagreements were settled via consensus. However, this tells us little about how often two skilled coders would come to similar conclusions. Either present information on reliability or list this as a limitation.

Authors’ response. As suggested by the reviewer, we have added information on reliability: An interrater reliability analysis using Cohen’s Kappa was performed for 20% of the coded interviews. The Kappa value was 0.81, which can be interpreted as “strong agreement” [1] (line numbers 170-172).

Comment 3. Also, at the bottom of page 13, the sentence that starts "the role of socializing patients' experiences with cancer..." does not make sense to me. What does it mean to "socialize" experiences? Please clarify or cut.

Authors’ response. Thank you for pointing this out. We have clarified what we meant by ‘socialize’ by editing the text at the bottom of page 13. The sentence now starts “the role of accessing patients’ experiences with cancer…” (line number 317). The word ‘socialize’ was also used in another sentence that we have also edited (line numbers 341-343):

The potential role of having access to these experiences in not just helping individuals seek medical attention, but also dealing with information and emotional aspects of a prospective cancer diagnosis at the beginning of their journey requires further investigation. 

References:

1. McHugh ML. Interrater reliability: The kappa statistic. Biochemia Medica. 2012;22(3):276-82. PubMed PMID: 82559661.

---

## [Editor Report · Decision Letter 2]

7 Sep 2020

From symptom to cancer diagnosis: Perspectives of patients and family members in Alberta, Canada

PONE-D-20-12776R2

Dear Dr. Pujadas Botey,

We’re pleased to inform you that your manuscript has been judged scientifically suitable for publication and will be formally accepted for publication once it meets all outstanding technical requirements.

Kind regards,

Alvaro Galli

Academic Editor

PLOS ONE

---

## [Editor Report · Acceptance letter]

14 Sep 2020

PONE-D-20-12776R2 

From symptom to cancer diagnosis: Perspectives of patients and family members in Alberta, Canada 

Dear Dr. Pujadas Botey:

I'm pleased to inform you that your manuscript has been deemed suitable for publication in PLOS ONE. Congratulations! Your manuscript is now with our production department. 

Kind regards, 

on behalf of

Dr. Alvaro Galli 

Academic Editor

PLOS ONE